# In Vitro Evaluation of Chemically Analyzed *Hypericum Triquetrifolium* Extract Efficacy in Apoptosis Induction and Cell Cycle Arrest of the HCT-116 Colon Cancer Cell Line

**DOI:** 10.3390/molecules24224139

**Published:** 2019-11-15

**Authors:** Shahinaz Mahajna, Sleman Kadan, Zipora Tietel, Bashar Saad, Said Khasib, Aziz Tumeh, Doron Ginsberg, Hilal Zaid

**Affiliations:** 1Qasemi Research Center, Al-Qasemi Academic College, P.O. Box 124, Baqa El-Gharbia 30100, Israel; 2The Mina and Everard Goodman Faculty of Life Science, Bar Ilan University, Ramat Gan 5299, Israel; 3Casali Center for Applied Chemistry, Institute of Chemistry, The Hebrew University of Jerusalem, Givat Ram, Jerusalem 91904, Israel; 4Department of Postharvest Science of Fresh Produce, ARO, The Volcani Center, Bet Dagan 7505101, Israel; 5Faculty of Sciences, Arab American University, P.O. Box 240, Jenin 009704, Palestine

**Keywords:** apoptosis, colon cancer, cell cycle, phytochemicals, *Hypericum triquetrifolium*

## Abstract

Naturally derived drugs and plant-based products are attractive commodities that are being explored for cancer treatment. This in vitro study aimed to investigate the role of *Hypericum triquetrifolium* (50% ethanol: 50% water) extract (HTE) treatment on apoptosis, cell cycle modulation, and cell cycle arrest in human colon cancer cell line (HCT-116). HTE induced cell death via an apoptotic process, as assayed by an Annexin V-Cy3 assay. Exposing HCT-116 cells to 0.064, 0.125, 0.25, and 0.5 mg/mL of HTE for 24 h led to 50 ± 9%, 71.6 ± 8%, 85 ± 5%, and 96 ± 1.5% apoptotic cells, respectively. HCT-116 cells treated with 0.25 and 0.5 mg/mL HTE for 3 h resulted in 38.9 ± 1.5% and 57.2 ± 3% cleavage of caspase-3-specific substrate, respectively. RT-PCR analysis revealed that the HTE extract had no effect on mRNA levels of Apaf-1 and NOXA. Moreover, the addition of 0.125 mg/mL and 0.25 mg/mL HTE for 24 h was clearly shown to attenuate the cell cycle progression machinery in HCT-116 cells. GC/MS analysis of the extract identified 21 phytochemicals that are known as apoptosis inducers and cell cycle arrest agents. All the compounds detected are novel in *H. triquetrifolium*. These results suggest that HTE-induced apoptosis of human colon cells is mediated primarily through the caspase-dependent pathway. Thus, HTE appears to be a potent therapeutic agent for colon cancer treatment.

## 1. Introduction

Cancer is the second leading cause of mortality in humans worldwide. It is estimated that one in three women and one in two men in the United States will develop cancer in their lifetime. An increase in the number of individuals diagnosed with cancer each year, Due to population growth and a higher life expectancy, there has been an increase in the number of individuals diagnosed with cancer each year, but, coupled with improving survival rates, the number of cancer-survivors has increased as well [1,2]. Colorectal cancer (CRC) is the second leading cause of cancer death. The development of CRC proceeds through one or a combination of three different mechanisms: chromosomal instability, epigenetic instability pathway named CpG island methylator phenotype, and microsatellite instability. Chromosomal instability pathway begins with the loss in functionality of the Adenomatous Polyposis Coli (APC) gene genes are italicized, which is a tumor suppressor gene [3]. This is followed by the inactivation of the p53 gene (a sensor essential for the checkpoint control that arrests cells with damaged DNA in the G1 phase), which results in the formation of polyps on the inside of the colon wall. Although much has yet to be understood as to why some individuals develop CRC while others do not, certain genetic and environmental factors are known to increase a person’s chance of developing the disease [1].

Surgery, chemotherapy, and radiotherapy, either individually or combined, were considered as conventional strategies for cancer treatment in the last century. With the rapid development of molecular medicine, novel therapeutic approaches such as immunotherapy, molecular targeted therapy, and hormonal therapy have been proposed to improve clinical outcomes for cancer patients. However, these therapeutic approaches are not always effective and survival rates are still poor [1,2]

There has been a substantial increase in the use of complementary and alternative medicines, including dietary supplements and medicinal plants, for cancer treatment. Several in vitro, cellular, and animal studies have examined the effects of herbal and other specialty products on the development and progression of CRC [3,4,5].

The use of agents targeting the cell cycle machinery has long been considered an ideal strategy for cancer therapy. These drugs target the abnormal expression of cyclin-dependent kinases (CDKs), mitotic kinases/kinesins, or they affect cellular checkpoints, resulting in cell cycle arrest and the subsequent induction of apoptosis in cancer cells. Cell-cycle-based agents can be grouped into categories that reflect their molecular targets. For example, CDK inhibitors target the inhibition of CDKs, which selectively block tumor growth without compromising normal cells; checkpoint inhibitors target the S and G2 checkpoints, and mitotic inhibitors affect mitosis [6,7,8]. Another key player in the cyclins-CDKs assembly is retinoblastoma protein (Rb). Phosphorylated Rb enhances E2F-1 to translocate into the nucleus and transcribe genes needed to switch the cell from G1 to S phase [9,10]. Indeed, overexpression of E2F-1, CDKs and cyclins in carcinoma are associated with the progression of metastasis [11,12,13]. Moreover, Natural and synthetic drugs were reported to prevent colon cancer cell growth by inhibiting the expression of key cell cycle regulating proteins such as, CDK2, cyclins E2F-1 [10,14,15].

Apoptosis induction is a useful mechanism for modulating cancer progression, especially when there are mutations that alter the ability of the cell to undergo apoptosis and allow transformed cells to keep proliferating rather than dying. It would be therapeutically advantageous to tip the balance in favor of apoptosis over mitosis in tumors, if possible. The progressive accumulation of genetic alterations (APC, p53, and ras) governs the transition of the normal colorectal epithelium to adenocarcinomas [16].

Herbal medicines such as garlic, onion, black fennel seeds, olive oil, olive leaves, as well as HTE, are prescribed for cancer treatment and prevention [17,18]. *Hypericum triquetrifolium* is also commonly used to treat inflammation, septic shock depression and nociception [19]. In addition, several studies have reported the efficacy of *Hypericum* genus in apoptosis induction in several cancer cell lines. For instance by active compounds from *Hypericum ascyron* L was effective in apoptosis induction in in human cervical carcinoma HeLa cells [20]. Intriguingly, *Hypericum elodeoides* augmented apoptosis in HeLa Cell through Caspase-8 Activation and PARP Cleavage [21]. Cisplatin and the hypericin found in *Hypericum perforatum* exhibited a dose-dependent cytotoxic and apoptotic effect in the MCF-7 cell line [22]. Moreover, *Hypericum adenotrichum Spach* [23], *Hypericum scabrum* L. [24] and *Hypericum japonicum* [25] induced apoptosis in several cancer cell lines *in vitro.* Despite the progress of modern medicine, traditional medicine is still being practiced [26]. However, the safety and effectiveness of alternative medicine are not always scientifically proven. Based on the knowledge of traditional herbal medicine and on preliminary studies, this in vitro study aimed to investigate the role of *Hypericum triquetrifolium* (50% ethanol: 50% water) extract (HTE) treatment on apoptosis, cell cycle modulation, and cell cycle arrest in a human colon cancer cell line (HCT-116).

## 2. Materials and Methods

### 2.1. Materials

Cells of the human colorectal cell line HCT-116 (ATCC^®^ CCL-247™) were purchased from the ATCC (Manassas, VA, USA). All tissue culture reagents, including fetal bovine serum and standard culture medium RPMI-1640, were purchased from Biological Industries (Beit Haemek, Israel). An LDH kit was purchased from Promega (Madison, WI, USA); a cell cycle kit was purchased from Thermo Fisher (Waltham, MA, USA); and an RNA isolation kit was purchased from QIAGEN (Hilden, Germany). MTT reagent and all other materials were purchased from Sigma Aldrich (St. Louis, MO, USA). HTE (aerial parts) was purchased from Al Alim—Medicinal Herb Center, Zippori, Israel.

### 2.2. Preparation of Plant Extracts

One hundred g of air-dried HTE plant material was added to 1 L of 50% EtOH (in water) and boiled for 30 min, then stirred for 24 h at room temperature. The obtained extract was filtered through filter paper, aliquoted, and frozen at −80 °C until use [27,28].

### 2.3. Silylation Derivatization

The dried extract sample was re-dissolved and derivatized in a solution of 40 μL of 20 mg/mL methoxyamine hydrochloride in pyridine for 90 min at 370 °C, followed by a 30 min treatment with 70 μL of N-methyl-N(trimethylsilyl)trifluoroacetamide (MSTFA) at 37 °C and centrifugation. 1 μL of the derivatized sample was injected into the gas chromatograph coupled with a mass selective detector (GC/MS) [29].

### 2.4. Gas Chromatography-Mass Spectrometry Analysis

The GC/MS system was comprised of a COMBI PAL autosampler (CTC Analytics, Zwingen, Switzerland), a Trace GC Ultra gas chromatograph equipped with a programmed temperature vaporizer (PTV) injector, and a DSQ quadrupole mass spectrometer (ThermoElectron Cooperation, Austin, TX, USA). GC analysis was performed on a Zebron ZB-5ms (30 m × 0.25 mm × 0.25 μm) MS column (Phenomenex, Torrance, CA, USA). The PTV split technique was carried out as follows: sugars were analyzed with a split of 1:100, and the lower abundance compounds were analyzed with a split of 1:10. The following temperature program was employed: the PTV inlet temperature was set to 45 °C and held there for 0.05 min, followed by a temperature increase to 70 °C, with a ramp rate of 10 °C/s. The PTV inlet was held at 70 °C for 0.25 min, followed by a transfer-to-column stage, whereby the temperature was increased to 270 °C, with a ramp rate of 14.5 °C/s, and held there for 0.8 min. Finally, a cleaning stage was carried out by raising the temperature to 330 °C, with a ramp rate of 10 °C/s, and maintaining it for 10 min.

The interface was heated to 300 °C, and the ion source was adjusted to 250 °C. Helium was used as the carrier gas at a flow rate of 1.2 mL/min. The analysis was performed using the following temperature program: 1 min isothermal at 40 °C, followed by an increase to 320 °C at a 15 °C/min ramp rate, and then maintaining this temperature for 4.5 min. Mass spectra were recorded at 9 scans/s in the *m*/*z* 40–450 scanning range from 5 till 10 min, and in the *m*/*z* 50–600 scanning range from 10 till 24 min. For the analysis of the lower abundance compounds, the filament was switched off from 12.95 to 13.60 min in order to prevent damage to the MS detector from the high concentration of sugar compounds [29].

### 2.5. Identification of Components

Phytochemical compounds were putatively identified by a correlation of their retention index (RI) and mass spectrum to those present in the mass spectra library of the Max-Planck-Institute for Plant Physiology (Golm, Germany; Q_MSRI_ID, http://csbdb.mpimp-golm.mpg.de/csbdb/gmd/msri/gmd_msri.html) and the commercial mass spectra library NIST05 (http://www.nist.gov/). The response values for metabolites resulting from the Xcalibur processing method were normalized to the ribitol internal standard [29,30].

### 2.6. Cell Culture

Human colorectal cell line HCT-116 were grown in Roswell Park Memorial Institute (RPMI)-1640 medium with a high glucose content (4.5 g/L) and supplemented with: 10% *v*/*v* fetal calf serum, 1% nonessential amino acid, 1% glutamine, and 100 U/mL penicillin. The cells were maintained in a humidified atmosphere of 5% CO_2_ at 37 °C.

### 2.7. MTT (Cell Viability)

The tetrazolium dye MTT method is a colorimetric assay based on the conversion of the yellow tetrazolium bromide to its purple formazan derivative by mitochondrial succinate dehydrogenase in viable cells. 20,000 cells were seeded per well of 96-microtiter plates. 24 h after cell seeding, the cells were incubated with increasing concentrations of plant extract (0−1000 µg/mL) for 24 h at 37 °C. The cells were then washed in phosphate-buffered saline and incubated in serum-free RPMI. After this, 0.5 mg/mL MTT was added to each well (100 µL) and incubated for 4 h. Afterward, the medium was removed, and the cells were incubated for 15 min with 100 µL of acidic isopropanol (0.08 N HCl), in order to dissolve the formazan crystals. The absorbance of the MTT formazan was measured at 570 nm in a microtiterplate reader. Viability was defined as the ratio (expressed as a percentage) of absorbance of treated cells to untreated cells [31].

### 2.8. Lactate Dehydrogenase

Lactate dehydrogenase (LDH) assay measures the leakage of the LDH enzyme, which is normally found in the cytoplasm, into the extracellular medium as an indication for plasma membrane rupture. LDH activity was measured in both the supernatants and the cell lysate fractions using CytoTox 96, a non-radioactive cytotoxicity assay kit (Promega), in accordance with the manufacturer’s instructions. The absorbance was measured at 490 nm with a 96-well microtiterplate reader (Anthos, Biochrom, Cambridge, UK) [31].

### 2.9. Apoptosis Detection

HCT-116 cells were seeded in a 24-well plate and incubated with HTE (0−500 µg/mL) for 24 h, followed by apoptosis detection using Acridine Orange (green) and Annexin-V CY3 (red) dyes, according to the manufacture’s protocol. Annexin V-Cy3 Apoptosis Detection Kit (abcam, Cambridge, UK) is based on the observation that soon after initiating apoptosis, cells translocate the membrane phospholipid phosphatidylserine (PS) from the inner face of the plasma membrane to the cell surface. Once at the cell surface, PS can be easily detected by staining with a fluorescent conjugate of Annexin V, a protein that has a high affinity for PS. The detection of the stained PS was carried out by fluorescence microscopy [32].

### 2.10. Caspase-3 Activity Assay

HCT-116 cells were treated as described in Section 2.9 and caspase-3 activity was detected according to the manufacturer’s protocol. Caspase-3 Colorimetric assay (Sigma-Aldrich) is based on the hydrolysis of the peptide substrate acetyl-Asp-Glu-Val-Asp p-nitroanilide (Ac-DEVD-pNA) by caspase-3, which results in the release of the *p*-nitroaniline (pNA) moiety. The concentration of pNA was calculated from the absorbance measured at 405 nm with a 96-well microtiterplate reader [33].

### 2.11. Cell Cycle Assay

HCT-116 cells were treated as described in Section 2.9. The cells were stained with a fluorescence ubiquitination cell cycle indicator (FUCCI, Premo FUCCI Cell Cycle Sensor, BacMam 2.0) according to the manufacturer’s instructions, in order to detect the cell cycle stages. This indicator employs red (RFP) and green (GFP) fluorescent proteins fused to different regulators of the cell cycle: Cdt1 and geminin. In the G1 phase of the cell cycle, only Cdt1 tagged with RFP may be visualized, thus identifying cells in the G1 phase with red fluorescent nuclei. In the S, G2 and M phases, Cdt1 is degraded and only geminin tagged with GFP remains, thus identifying cells in these phases with green fluorescent nuclei. During the G1/S transition, both proteins are present in the cells, allowing GFP and RFP fluorescence to be observed together as yellow fluorescence. The cells monitored under fluorescent microscope [34].

### 2.12. FACS Analysis

HCT-117 cells were seeded in a 24-well plate and a day after they were treated with the plant extract (0−250 µg/mL) for 24 h, trypsinized and fixed with 70% ethanol. After fixation, cells were centrifuged and incubated for 4 min at 1500 rpm at 4 °C. Then, the cells were suspended with 5 mg/mL propidium iodide and 50 μg/mL RNase A. After 20 min incubation at room temperature, fluorescence was measured using Becton Dickinson flow cytometer [35].

### 2.13. Total RNA Isolation and cDNA Synthesis

HCT-116 cells were treated as described in Section 2.9. Total RNA was isolated from the cells using Rneasy Plus Mini Kit (QIAGEN) according to the manufacturer’s instructions, and immediately frozen at −80 °C until further use. DNase-treated RNAs were used to synthesize cDNA with the Transcriptor First Strand cDNA Synthesis Kit, using random hexamers as specified by the manufacturer (Maxima First Strand cDNA Synthesis Kit for RT-PCR by Thermo).

Real-time PCR amplification (RT-PCR) and advanced relative quantification analysis were achieved using a Light Cycler 480 instrument (Roche Applied Science, Penzberg, Germany) with software version LCS480 1.5.039. All reactions were performed in duplicate with the Light Cycler Fast Start DNA Master SYBER Green I Kit (Roche Applied Science) in a final 20 µL volume with 2.5 mM MgCl_2_, 0.2 µM of each primer and 2 µL cDNA. Amplification conditions consisted of an initial pre-incubation step at 95 °C for 10 min (polymerase activation), followed by amplification of the target cDNA for 45 cycles (95 °C for 15 s, 60 °C for 20 s, and extension time at 72 °C for 30 s) [35]. the primer sequences (5′ to 3′) for qPCR amplification were:Apaf-1 for-AACCAGGATGGGTCACCATAApaf-1 rev-ACTGAAACCCAATGCACTCCNOXA for-CAGAGCTGGAAGTCGAGTGNOXA rev-CAGGTTCCTGAGCAGAAGAG

### 2.14. Statistical Analysis

Error limits were cited, and error bars were plotted and represent simple standard deviations of the mean. When comparing different samples, results were considered to be statistically different when *P* < 0.05. Student *t*-test was applied for statistical calculations using SPSS version 21.0 (Armonk, NY, USA)

## 3. Results Sections in Wrong Order—Experimental Is Last-Renumber Everythi8ngn

### 3.1. The Affects HCT-116 Cell Viability

The effect of HTE on cell viability was evaluated in the HCT-116 colon cancer cell line using MTT and LDH assays. Cells seeded in 96-well plates (20,000 cells/well) were exposed to increasing concentrations (0−1 mg/mL) of HTE for 24 h. Concentrations that led to less than 10% of cell death were considered non-toxic. HTE was determined to be non-toxic up to 0.5 mg/mL and the IC_50_ was ~1 mg/mL (Figure 1). HTE concentrations higher than 0.5 mg/mL caused a significant reduction in cell viability.

Membrane integrity can be evaluated by measuring LDH activity. LDH, an enzyme located in the cytoplasm, catalyzes the conversion of lactate to pyruvate. When LDH is found in the media of the cells, there are two possible causes: (1) cellular death or (2) a leak in the cell membrane. HCT-116 cells seeded in a 96-well plate (20,000 cells/well) were exposed to increasing concentrations (0−1 mg/mL) of HTE for 24 h. In accordance with the MTT results, no significant change in LDH levels was detected in the culture medium after exposure to extracts of HTE at concentrations of up to 0.25 mg/mL (Figure 2). Based on the MTT and LDH assay results, HTE was used in concentrations no greater than 0.5 mg/mL in the ensuing experiments.

### 3.2. HTE Induces Apoptosis in HCT-116 Cells

Annexin-V detects cells in early apoptosis stages via membrane-associated processes, by binding to PS head groups. An Annexin test was performed on HCT-116 cell line after exposure to 0.064, 0.125, 0.25, and 0.5 mg/mL of HTE. Characteristically, Annexin-V binds only to the surface of the HCT-116 cell membrane in apoptotic cells, where it binds to the exposed PS head groups [32]. As shown in Figure 3B, when compared with control treated cells, treatment of HCT-116 cells with HTE (0.125 and 0.25 mg/mL for 24 h) resulted in a significant increase of apoptosis (red membrane). Indeed, HTE induced 50 ± 9%, 71.6 ± 8%, 85 ± 5%, and 96 ± 1.5% apoptosis at concentrations of 0.064, 0.125, 0.25, and 0.5 mg/mL (Figure 3A,B). The putative apoptosis inducer structures and names are shown in Figure 3C. Most of the compounds are fatty acids and phenols as determined by GC/MS.

To further assess HTE’s ability to induce apoptosis, its effect on caspase-3 activation in HCT-116 cells was analyzed. As shown in Figure 4, treatment of HCT-116 cells with 0.25 and 0.5 mg/mL of HTE, for 3 h resulted in 38.9 ± 1.5% and 57.2 ± 3% cleavage of caspase-3-specific substrate, respectively. Staurosporine, which is commonly used as a positive control for apoptosis induction, (1 µM) led to 100% apoptosis. These results are in line with the previous Annexin V-Cy3 apoptosis detection results. Put together, these results indicate that HTE’s ability to induce apoptosis in HCT-116 cells is mediated, at least in part, through the activation of caspase-3.

### 3.3. mRNA Levels of Apaf-1 and NOXA

Next, the effect of HTE on the expression of apoptotic protease-activating factor-1 (Apaf-1) and NOXA, two proteins involved in the intrinsic apoptosis pathway, was examined. mRNA levels of Apaf-1 and NOXA were detected by RT-PCR and normalized to glyceraldehyde 3-phosphate dehydrogenase (GAPDH) levels. RT-PCR analysis showed HTE had no effect on Apaf-1 and NOXA mRNA levels (data not shown).

### 3.4. HTE Modulates the Cell Cycle

The effect of HTE on HCT-116 cell cycle was tested using FACS analysis. Cells were exposed to 0.125 and 0.25 mg/mL of HTE for 4, 8, and 24 h. Treatment of HCT-116 cells with HTE at concentrations of 0.125 and 0.25 mg/mL for 24 h resulted in a significantly higher level of sub G1 phase (21.7 ± 4.8% and 23.24 ± 3.9%, respectively), compared to non-treated cells (0.46 ± 0.22) as shown in Figure 5A,B.

No significant changes in the percentage of cells in the S and G2-M phases were observed. Treatments for 4 and 8 h with HTE did not result in a significant effect on the cell cycle distribution and cell viability. For further examination of the role HTE plays in cell cycle modulation, the FUCCI Cell Cycle Sensor assay was employed. HCT-116 cells challenged with 0.125 and 0.25 mg/mL HTE for 24 h were arrested in the G1 phase (red cells, Figure 5C). The putative cell cycle arresting compounds were found to be mainly terpenoids, phenols, and fatty acids as detected by GC/MS (Figure 5D). These compounds are also known as apoptotic inducers, except for dihydroxyacetone (Table 1).

### 3.5. GC-MS Analysis of the HTE

The phytochemical profile of HTE was complemented by GC/MS metabolite profiling of derivatized extracts, as described in Section 2.4. A total of 50 novel identified metabolites in *H. triquetrifolium* were monitored (Table 1). Most of the compounds were terpenoids, organic acids, alcohols, and sugars. Interestingly, 21 compounds detected are associated with either anti-cancer, apoptosis induction, or cell cycle arrest activity (Figure 6 and Table 1). These potential active compounds compose 23.2% of the total amount of the detected chemicals.

## 4. Discussion

The exploration of herbal medicines may present novel strategies for the treatment of CRC, which remains the second leading cause of cancer death. Herbal medicines such as garlic, onion, black fennel seeds, olive oil, olive leaves, and HTE are commonly prescribed for cancer treatment and prevention [17,18].

In this in vitro study, HTE was found to be non-toxic up to 0.5 mg/mL. Our study indicated that the treatment of a colon cancer cell line (HCT-116) with HTE extracts resulted in a significant induction of apoptosis, as detected by Annexin-v staining. The apoptotic effects of HTE were further confirmed by measuring caspase-3 activity. Caspases are a family of proteases that mediate cell death; they play an important role in the process of apoptosis. Activated caspase-3 is the key catalyst of the apoptosis process. Treatment of HCT-116 cells with HTE extract resulted in a dose-dependent activation of caspase-3. Based on these results, HTE-induction of apoptosis was determined to be mediated, at least partially, by caspase-3 activation.

The release of cytochrome c from the mitochondria serves as a trigger for the formation of apoptosome, an oligomeric protein complex consisting of Apaf1, procaspase-9, and cytochrome c [73]. The formation of apoptosome leads, in turn, to the activation of caspase-3. NOXA is a member of the bcl-2 family and is described as a p53 target gene, serving as a candidate mediator for p53-induced apoptosis [74]. Both NOXA and Apaf-1 genes were selected for further analysis of the underlying mechanisms of HTE-induced apoptosis. However, RT-PCR analysis revealed that HTE extracts had no effect on the mRNA levels of Apaf-1 and NOXA, suggesting that apoptosis induction by HTE is not governed by the transcriptional regulation of these genes. Apoptosis is closely regulated by anti-apoptotic and pro-apoptotic effector molecules and can be mediated by several distinct pathways. Gene expression is often interpreted in terms of protein levels. Production and maintenance of cellular proteins require a remarkable series of linked processes from transcription, processing, and degradation of mRNA to the translation, localization, modification, and programmed destruction of the proteins themselves [75]. Based on the results obtained, where HTE had no effect at the transcriptional level of the tested mRNAs, the induction of apoptosis by HTE extracts is, most probably, at a post-transcriptional level.

In addition, the effect of HTE extracts on the cell cycle progression was determined. The significant increase in the percentage of HCT-116 cells with a subG1 DNA content observed in the study suggests that this extract induced apoptosis and thereby disrupted the uncontrolled cell cycle progression.

GC/MS of the HTE extract was examined in order to detect bioactive compounds. Out of the 50 compounds detected, 21 compounds are associated with either anti-cancer, apoptosis induction, or cell cycle arrest activity. Most of the cell cycle arrest and apoptosis inducer compounds detected are phenols, terpenoids, fatty and organic acids, and alcohols. Indeed, plant extracts that contain phenolic compounds or long-chain fatty acids [76,77] have been reported as potential anti-cancer agents [78,79]. The specific activity of each compound detected is summarized in Table 1. Further studies are required in order to evaluate the anti-cancer activity of these compounds individually and to determine whether they are effective as pure compounds or combined.

Further studies are needed to map the different genes involved and the specific cellular pathways that influence the induction of apoptosis by HTE. Identifying and isolating agents that can induce apoptosis and cell cycle arrest in cancer cells is a high priority. HTE extract seems to possess a potent therapeutic activity for colon cancer via cell cycle arrest and apoptosis induction. However, HTE’s effect on untransformed cells needs to be further investigated before it can be used as an anti-cancer agent.

## Figures and Tables

**Figure 1 molecules-24-04139-f001:**
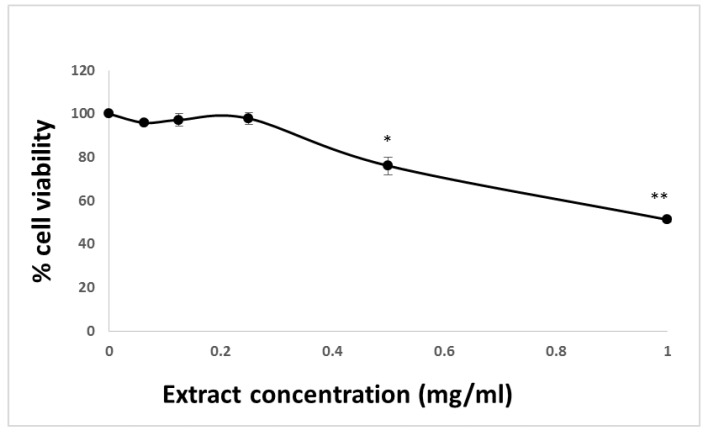
MTT assay of HCT-116 cells after 24 h treatment with varying concentrations of HTE. The absorbance of the MTT formazan was measured at 620 nm using microtiterplate reader. Cell viability was defined as the absorbance ratio (expressed as a percentage) of treated cells to untreated cells. Values represent the mean ±SD of three independent experiments carried out in triplicates. * *p* < 0.05, ** *p* < 0.01, significant as compared with controls.

**Figure 2 molecules-24-04139-f002:**
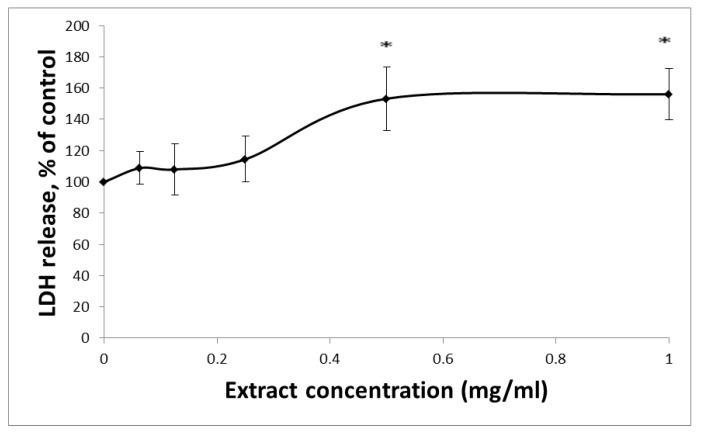
LDH leakage from HCT-116 cells after 24 h incubation with varying concentrations of HTE. The leakage of LDH into the extracellular medium is measured. The absorbance was measured at 492 nm using a microtiterplate reader. Values represent the mean ±SD of three independent experiments carried out in triplicates. * *p* < 0.05, significant as compared with controls.

**Figure 3 molecules-24-04139-f003:**
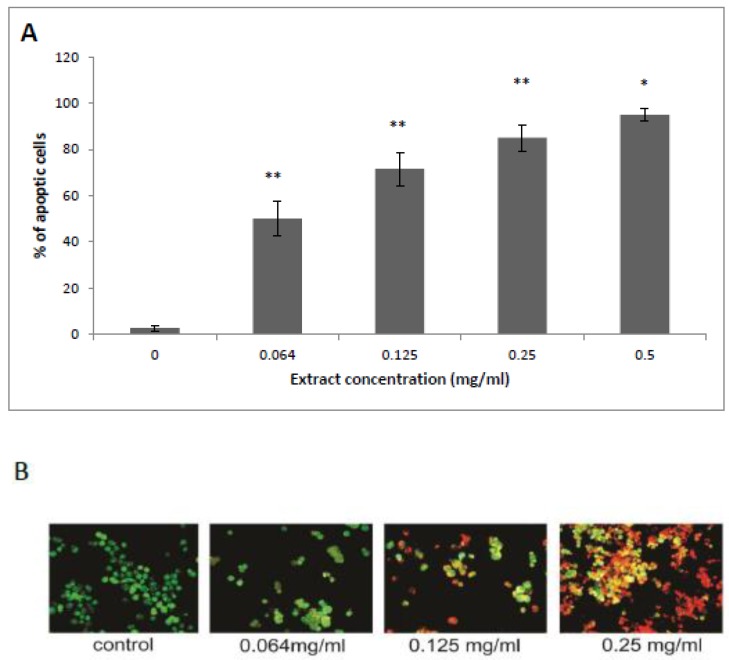
Determination of apoptosis induced by HTE in HCT-116 cells. Cells were exposed to the plant extract (up to 0.25 mg/mL) for 24 h. Apoptosis was determined using Acridine Orange (green) and Annexin-V CY3 (red) staining assay and was monitored by a fluorescence microscope. Apoptosis is expressed as a percentage of treated cells to untreated cells. (**A**) Values represent the mean ±SD of three independent experiments carried out in triplicates. * *p* < 0.05, ** *p* < 0.01, significant as compared with controls. (**B**) Representative fluorescence microscopy images showing co-staining with Acridine Orange and Annexin-V CY3. (**C**) Chemical structures of the detected putative apoptotic inducer compounds.

**Figure 4 molecules-24-04139-f004:**
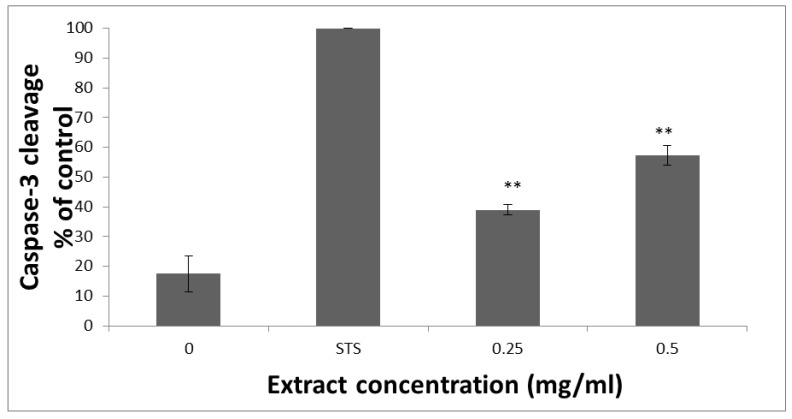
Analysis of intracellular caspase-3 activity in HCT-116 cells after 3 h post-treatment with staurosporine (1 µM) and HTE. Cell lysates were combined with the caspase-3-specific substrate in a standard reaction buffer. Cleavage of the caspase-3-specific substrate was compared to the staurosporine (STS, 1 µM) treated cells. The absorbance was measured at 405 nm using a microtiterplate reader. Values represent the mean ±SD of three independent experiments carried out in triplicates. ** *p* < 0.01—significant as compared with control.

**Figure 5 molecules-24-04139-f005:**
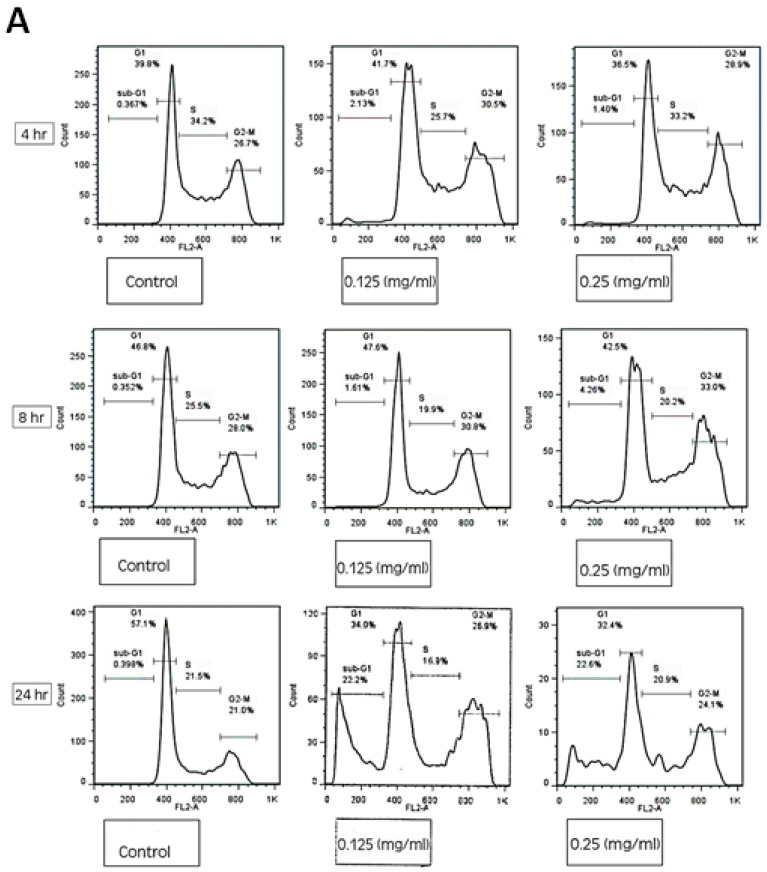
The effect of HTE on cell cycle distribution of HCT-116 cells. (**A**) Cellular DNA was stained with propidium iodide and flow cytometric analysis was done to determine the cell cycle distribution post-treatment with HTE for 4, 8, and 24 h. (**B**) The apoptotic fraction, the sub G1 phase of the cell cycle, is represented on the histograms. (**C**) Fluorescence images of HCT-116 cells using FUCCI Cell Cycle Sensor. Cells were treated with 0.125 and 0.25 mg/mL of HTE for 24 h. Three critical phases can be observed in FUCCI: G0/G1 (Red), G1/S (yellow), and G2/M (Green). (**D**) Chemical structure of the detected putative cell cycle arrest compounds.

**Figure 6 molecules-24-04139-f006:**
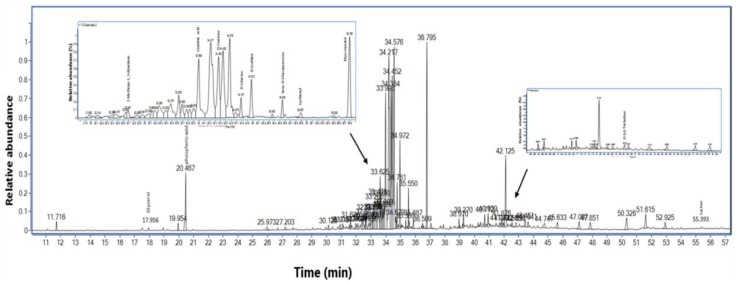
GC-MS chromatogram of the HTE. Major peaks are labeled with the compounds identified. Zoom; region of the elution of some compounds.

**Table 1 molecules-24-04139-t001:** Phytochemical profile of HTE and their association with anti-cancer, apoptosis induction, or cell cycle arrest activity.

Peak	Name	R_t_	% Area	DW mg/g	Match Factor	Relation to Apoptosis	References
1	Pentanoic acid	10.791	0.01	0.054207	89	Its derivatives are used to induce apoptosis in cell lines.	[36]
2	2,3-Butanediol	12.863	0.05	0.286145	98	N/A	
3	1,3-Propanediol	13.570	0.01	0.05732	88	N/A	
4	Lactic Acid	13.819	0.14	0.891773	93	N/A	
5	Hexanoic acid	14.002	0.04	0.276932	96	Also known as caproic acid. Reported to induce apoptosis in human colorectal, skin, and breast cancer. Moreover, it “could potentially be used to prevent and/or treat these cancers”.	[37]
6	Succinimide	15.525	0.01	0.060801	88	N/A	
7	Benzyl alcohol	16.447	0.11	0.667908	91	1% benzyl alcohol was reported to induce high apoptosis and necrosis in human dermal fibroblasts.	[38]
8	4-Hydroxybutanoic acid	16.96	0.07	0.465751	86	N/A	
9	Glycerol	17.956	6.27	39.74729	99	N/A	
10	3-Hydroxybutanoic acid	18.065	0.01	0.075168	92	N/A	
11	Dihydroxyacetone	18.768	0.03	0.184817	78	Reported to induce G2/M arrest and apoptotic cell death in melanoma A375P cell line.	[39]
12	Benzoic Acid	19.244	0.19	1.22039	98	Sodium benzoate was reported to activate NFκB and induce apoptosis in HCT116 cells.	[40]
13	Octanoic acid	19.808	0.02	0.143862	83	N/A	
14	phosphoric acid	20.467	3.85	24.42644	86	N/A	
15	1,2,3-Butanetriol	20.958	0.30	1.88296	98	N/A	
16	Nonanoic acid	22.371	0.11	0.679532	87	Reported to induce apoptosis in vivo in epidermal Langerhans.	[41]
17	(*E*)-Erythrono-1,4-lacton	22.971	0.04	0.230517	89	N/A	
18	Pyroglutamic acid	25.973	0.69	4.36759	94	N/A	
19	L-Threitol	26.515	0.09	0.581057	93	N/A	
20	meso-Erythritol	26.698	0.33	2.11151	95	N/A	
21	3-Hydroxybenzoic acid	27.584	0.11	0.674429	85	N/A	
22	4-Hydroxybenzoic acid	29.078	0.36	2.265056	81	Detected in *Schisandra chinensis* fruit extract, which is known to induce caspase-dependent apoptosis in human ovarian cancer A2780 cells. It was also detected in pinecones of *Pinus koraiensis* extract and exhibited cytotoxic activity, with IC_50_ value around 1 mg/mL in four human lung cancer cell lines, A549, H1264, H1299, and Calu-6. Similarly, Sitarek and colleagues reported antiproliferative and cell cycle arrest activity on glioma cells for the same compound in the root extract of *Leonurus sibiricus L.* It increased Bax, Bcl-2, p53, Caspases in glioma and breast cancer cell lines.	[42,43,44,45]
23	Arabinonic acid, γ-lactone	29.429	0.06	0.402076	89	N/A	
24	Phloroglucinol	29.59	0.09	0.590668	92	Reported to induce apoptosis in distinct cancer cell lines.	[46,47,48]
25	Levoglucosan	31.084	0.66	4.189809	95	N/A	
26	d-(-)-Rhamnose	31.545	0.31	1.948759	94	Apoptosis inducer and anti-cancer agent, especially in human breast cancer.	[49]
27	Vanillic Acid	31.962	0.71	4.500047	77	An antioxidant and has some anti-cancer benefits.	[50]
28	2-Methoxy-1,3-dioxolane	32.687	2.80	17.73789	91	N/A	
29	Methyl α-d-glucofuranoside	32.951	0.52	3.281267	86	Close derivates are used for tumor treatment.	
30	Protocatechuic acid	33.068	0.25	1.559802	84	Known to induce apoptosis in human ovarian, breast, lung, liver, cervix, and prostate cancer cells, as well as others, by modulating FAK, MAPK, c-June, and NF-κB pathways.It was also reported to lead to cell cycle arrest.	[51,52,53,54,55]
31	Shikimic acid	33.068	0.46	2.892089	87	N/A	
32	Quininic acid	33.998	13.41	85.04314	88	N/A	
33	D-Fructose	34.364	18.78	119.1289	91	N/A	
34	*p*-Coumaric acid	34.679	0.24	1.521807	89	Reported to induced apoptosis and cell cycle arrest in several cell lines, including human colon cancer.	[56,57,58]
35	D-Glucose	34.781	3.34	21.1588	96	N/A	
36	D-Sorbitol	34.972	6.79	43.05749	94	N/A	
37	β-d-Glucopyranose	35.55	2.72	17.22668	98	N/A	
38	3-Hexyl-7,8,9,10-tetrahydro-6,6,9-trimethyl-6*H*-dibenzo(b,d)pyran-1-ol,	35.887	1.07	6.810763	79	Also called Synhexyl	
39	Palmitic Acid	35.887	1.07	6.810763	98	Reported to induce apoptosis in dozens of cancer cell lines via MAPK and AMPK/Akt/mTOR, miR-129-3p/Smad3, and estrogen receptor alpha signaling pathways.	[59,60,61]
40	Myoinositol	36.795	18.69	118.5457	97	Reported to be involved in apoptosis induction in the Arabidopsis plant.	[62]
41	Caffeic acid	37.066	0.50	3.155426	81	Reported to induce apoptosis and cell cycle arrest in several cell lines, including human colon, breast, nasopharyngeal carcinoma, melanoma, lung, nasopharyngeal and others. It altered the mTOR/PI3K/AKT signaling pathway and inactivated NF-κB pathway.	[63,64,65]
42	L-Rhamnose	37.74	0.26	1.662177	85	N/A	
43	Stearic acid	37.915	0.52	3.312062	91	Reported to arrest the cell cycle and induce apoptosis in HepG2 and other cancer cell lines.	[66,67]
44	Glyceryl-glycoside	38.97	0.72	4.565657	94	N/A	
45	D-(+)-Trehalose	43.451	1.31	8.317046	92	Induced autophagy.	[68]
46	(2*R*)*-*(*E*)-Catechine	44.747	0.55	3.465693	86	N/A	
47	(2*R*-*cis*)-Catechine	45.142	0.08	0.496909	84	N/A	
48	β-Sitosterol	53.782	0.19	1.17753	88	Also named phytosterol. Reported to induce apoptosis and cell cycle arrest in HCT116, MCF-7, A549, and HeLa cell lines. Altered the PI3K/Akt signaling pathway and AMPK/PTEN/HSP90.	[69,70,71]
49	Stigmasterol	53.782	0.18	1.15534	90	Reported to lead to cell cycle arrest, mitochondrial-mediated apoptosis, and inhibition of JAK/STAT signaling pathway. It also inhibited cell migration in human gastric cancer cells.	[72]
50	Sucrose	55.393	10.91285	69.21846	92	N/A	

N/A, not applicable.

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
