# Peer review of "In Vitro Evaluation of Chemically Analyzed Hypericum Triquetrifolium Extract Efficacy in Apoptosis Induction and Cell Cycle Arrest of the HCT-116 Colon Cancer Cell Line"

_molecules, 2019, doi:10.3390/molecules24224139_

Round 1

Reviewer 1 Report

Title: The title should be rephrased or reworked. It should be concise and easily followed. As is, it requires the reader to read it over and over before understanding what is being reported. Additionally, the name of the cell line must be reflected.

Abstract: the Caspase-3 data should be explicitly mentioned in the abstract.

Some of the data in the abstract could be put in a better way. The authors must consider revising the abstract. Statistical analysis is not included.

Introduction: The authors forgot to introduce the plant and why it was important to investigate its potential apoptotic and cell cycle modulatory effects. There are several papers that have reported the biological activities of the Hypericum genus. There is also a paper on the activity of the plant under this genus against different cell lines, including colon cancer cell lines.

Line 42-43: The cause of colon carcinogenesis is rather controversial. The authors write as if this is the only cause of colon cancer. I would also suggest that the authors also cover briefly what causes cancer. Line 46 - 48 is safer and sensible.

Line 56: The authors talk about several in vitro, cellular and and animal studies..... But they cite one article. this is a sign of cross-referencing. 

Line 57: Special or Specialty?

Line/s 59 - 65: The authors need to link the CDKs to the carcinogenesis process and provide examples, especially in colon cancer.

Methods:

The extract has not been tested in non-cancer cells, either liver or kidney cells to ascertain safety.

Line 81: ATCC number must be supplied with the cell line name not with the company. For example: HCT-116 cell line (ATCC CCL-247) was purchased from ATCC, USA.

Line 126: The first sentence needs attention.

Line 134: how many extracts?

Line 139: Is it Elisa reader or plate reader? Provide the actual name of the equipment and the supplier.

2.11. Cell Cycle Assay: This technique is not clear, especially how the visualization was done.

Which tests were used to determine statistical significance?

Results

The subtitles should reflect the findings.

Figure 1: Y axis label should read % Cell Viability.

Figure 2: Y axis label % of control does not say much.

Positive control data is missing, it is only shown in figure 4.

Solvent control data is also missing.

Y axis label "% of control" does not reveal much.

There is data in the literature that show that other Hypericum sp. regulated Bax, Bcl-2 and p53. Why didn't the authors investigate these? Or even look at the protein level?

In figure 5, the cell cycle data should have been analysed statistically and represented quantitatively.

There is a lot of compounds that were identified in this study, were there novel compounds detected in this plant?

In table 1, N/A is not defined. The table contains interesting information of the regulatory role of some of the compounds identified in this study. it would have been informative if some of the reported genes and pathways were investigated.

Discussion:

The authors only looked at apaf-1 and NOXA (data not shown) but claimed that apoptosis induction is not regulated at the transcriptional level. p53, bcl-2 and bax should have been looked at, as well as other genes reported in table 1. 

Author Response

 (x) Extensive editing of English language and style required 

The paper was revised by the Academic Language Experts. The specific person who edited the manuscript is a native English and expert in scientific papers editing. Kindly find the uploaded certificate.

Title: The title should be rephrased or reworked. It should be concise and easily followed. As is, it requires the reader to read it over and over before understanding what is being reported. Additionally, the name of the cell line must be reflected.

The title was changed to: In vitro evaluation of chemically analyzed H.triquetrifolium extract efficacy in apoptosis induction and cell cycle arrest of HCT-116 colon cancer cell line

Abstract:

The Caspase-3 data should be explicitly mentioned in the abstract.Some of the data in the abstract could be put in a better way. The authors must consider revising the abstract. Statistical analysis is not included.

The abstract is revised now and we added the statistics.

Introduction: The authors forgot to introduce the plant and why it was important to investigate its potential apoptotic and cell cycle modulatory effects. There are several papers that have reported the biological activities of the Hypericum genus. There is also a paper on the activity of the plant under this genus against different cell lines, including colon cancer cell lines.

We added new paragraph (line 89-97).

Line 42-43: The cause of colon carcinogenesis is rather controversial. The authors write as if this is the only cause of colon cancer. I would also suggest that the authors also cover briefly what causes cancer. Line 46 - 48 is safer and sensible.

The paragraph “The development of CRC proceeds through a series of genetic alterations involving the activation of oncogenes and the loss of tumor suppressor genes. The first step in colon carcinogenesis involves the loss in functionality of the APC (Adenomatous Polyposis Coli) gene, which is a tumor suppressor gene.”

Was substituted by:

The development of CRC proceeds through one or a combination of three different mechanisms: chromosomal instability, epigenetic instability pathway named CpG island methylator phenotype, and microsatellite instability. Chromosomal instability pathway begins with the loss in functionality of the APC (Adenomatous Polyposis Coli) gene, which is a tumor suppressor gene (reference in the manuscript)

Line 56: The authors talk about several in vitro, cellular and and animal studies..... But they cite one article. this is a sign of cross-referencing.

We added additional two references.

Line 57: Special or Specialty?

It is specialty products meaning supplementary products.

Line/s 59 - 65: The authors need to link the CDKs to the carcinogenesis process and provide examples, especially in colon cancer.

We added explanation as required (lines 74-80)

Methods:

The extract has not been tested in non-cancer cells, either liver or kidney cells to ascertain safety.

The extract safety was tested by MTT and LDH leakage assay in HCT-116 cells.  We did not include other cell type in this study as we focused only on HCT-116 cells. Our aim was to point out potent active compounds in Hypericum triquetrifolium extract and the potential target in HCT116 cells.

Line 81: ATCC number must be supplied with the cell line name not with the company. For example: HCT-116 cell line (ATCC CCL-247) was purchased from ATCC, USA.

Corrected accordingly.

Line 126: The first sentence needs attention.

Corrected, thanks.

Line 134: how many extracts?

Extracts corrected to extract.

Line 139: Is it Elisa reader or plate reader? Provide the actual name of the equipment and the supplier.

Thank you for your comment. It is microtiterplate reader (Anthos). We updated the manuscript.

2.11. Cell Cycle Assay: This technique is not clear, especially how the visualization was done.

We added now “The cells monitored under fluorescent microscope”.

Which tests were used to determine statistical significance?

We used student t-test. The methods are updated now.

Results

The subtitles should reflect the findings.

Subtitles are modified now.

Figure 1: Y axis label should read % Cell Viability.

We changed it to: % Cell viability

Figure 2: Y axis label % of control does not say much.

We changed it to: LDH release, % of control

Positive control data is missing, it is only shown in figure 4.

The results in figure 1&2 are presented as a percent of the alive cells. Therefore, the positive control may not be applicable in such curve. In figure 3, apoptosis induction reached almost 100% when the cells were treated with 0.5mg/ml HTE. In figure 5 the shift in the cell cycle is compared relative to untreated cells.

Solvent control data is also missing.

All the negative controls are indeed vehicle. All the cells treated with 0mg/ml HTE, were treated with ethanol (equal to the amount in the highest HTE concentration).

Y axis label "% of control" does not reveal much.

Changed to “Caspase-3 cleavage, % of control”

There is data in the literature that show that other Hypericum sp. regulated Bax, Bcl-2 and p53. Why didn't the authors investigate these? Or even look at the protein level?

This study does not focus on the gene expression. Yet, we thought to test the effect of HTE on two key genes in the apoptosis pathway: NOXA is a member of the bcl-2 family and a candidate mediator for p53-induced apoptosis; and Apaf-1 needed for the apoptosome formation.

In figure 5, the cell cycle data should have been analysed statistically and represented quantitatively.

We added now the statistical analysis (Fig. 5B) for the cell cycle distribution as required.

There is a lot of compounds that were identified in this study, were there novel compounds detected in this plant?

All the compounds identified here in Hypericum triquetrifolium were not mentioned elsewhere in this plant.

In table 1, N/A is not defined.

Added now below the table.

The table contains interesting information of the regulatory role of some of the compounds identified in this study. it would have been informative if some of the reported genes and pathways were investigated.

We agree with the reviewer. This is very interesting point and we are planning to proceed in this project.  Indeed, we asked for additional fund to study specific pathways reported in the table and investigate the synergism and antagonism of some key detected compounds found in the extract.

Discussion:

The authors only looked at apaf-1 and NOXA (data not shown) but claimed that apoptosis induction is not regulated at the transcriptional level. p53, bcl-2 and bax should have been looked at, as well as other genes reported in table 1.

Gene expression is beyond the scoop of this manuscript. As we mentioned above, we are now preparing for new funded project to try addressing the questions raised here and to test the precise effect of the key detected compounds listed in the table. Our conclusion was: apoptosis induction by HTE is not governed by the transcriptional regulation of these genes.

Reviewer 2 Report

Mahajna and Kadan et al. presented the H. triquetrifolium extract (HTE) exhibited cytotoxicity to colon cancer cell line HCT116 by inducing cell cycle arrest and apoptosis. The authors analyzed the components of THE by GC/MS and revealed several potent bioactive compounds might cause the cytotoxicity. Generally, this work is showing that HTE actively inhibited CRC cells and can be improved before publishing.

Major points

The authors demonstrated that THE induced apoptotic cell death by PS staining. The molecular examination is required to confirm this effect. Detection of protein expression level of apoptosis markers such as cleavage of PARP and caspase 3/7, BAX, and BCL2 is highly recommended. How many times did the author examine the cell cycle analysis? What’s the reason causing inconsistence in the results of cell cycle distribution analysis and FUCCI cell cycle sensor assay? Statistical analysis of cell cycle distribution is required to convince the readers whether the result is significant or not. Markers for cell cycle gap keepers is required to check the effects of HTE on cell cycle arrest.

Minor points

The title need to be revised to be concise and specific. I suggest the authors to include a paragraph to introduce relevant studies of triquetrifolium. I suggest the authors to move the result of “3.3 mRNA Levels of Apaf-1 and NOXA” and method of “2.13 Total RNA Isolation and cDNA Synthesis” to supplementary data and provide the primer sequences for qPCR amplification.

Author Response

Comments and Suggestions for Authors

Mahajna and Kadan et al. presented the H. triquetrifolium extract (HTE) exhibited cytotoxicity to colon cancer cell line HCT116 by inducing cell cycle arrest and apoptosis. The authors analyzed the components of THE by GC/MS and revealed several potent bioactive compounds might cause the cytotoxicity. Generally, this work is showing that HTE actively inhibited CRC cells and can be improved before publishing.

Major points

The authors demonstrated that THE induced apoptotic cell death by PS staining. The molecular examination is required to confirm this effect. Detection of protein expression level of apoptosis markers such as cleavage of PARP and caspase 3/7, BAX, and BCL2 is highly recommended.

We would like to thank you for the valuable suggestions. We agree with the reviewer. This is very interesting point and we are planning to proceed in this project. Indeed, we had applied for new fund to study specific pathways reported in this paper especially the effect of HTE and active ingredients reported in the table on the apoptosis proteins markers.

How many times did the author examine the cell cycle analysis? What’s the reason causing inconsistence in the results of cell cycle distribution analysis and FUCCI cell cycle sensor assay? Statistical analysis of cell cycle distribution is required to convince the readers whether the result is significant or not

All the experiments in this project were performed at least 3 times in this project including FACS. We added now the statistical analysis (Fig. 5B) for the cell cycle distribution as required. The effect of HTE on HCT-116 cell cycle was tested using FACS analysis and FUCCI. HTE led to significantly higher level of sub G1 phase as detected by FACS. HCT-116 cells challenged with the same concentration and time with HTE (as in the FACS experiment) were arrested in the G1 phase. The two results are much closed, taking into account that there is no specific dye for sub G1 phase in the FUCCI staining.

Markers for cell cycle gap keepers is required to check the effects of HTE on cell cycle arrest.

The assay was applied using the cell cycle indicator (FUCCI, Premo FUCCI Cell Cycle Sensor, BacMam 2.0) according to the manufacturer’s instructions. We could not find specific markers for cell cycle keepers in the kit used.

Minor points

The title need to be revised to be concise and specific. I suggest the authors to include a paragraph to introduce relevant studies of triquetrifolium.

Thanks for the suggestion. Done

I suggest the authors to move the result of “3.3 mRNA Levels of Apaf-1 and NOXA” and method of “2.13 Total RNA Isolation and cDNA Synthesis” to supplementary data and provide the primer sequences for qPCR amplification.

We added the primers sequence in the methods section. Since other reviewers asked to add more explanation regarding this result, we thought to leave it in the main manuscript.

Reviewer 3 Report

In the manuscript, the authors investigated the role of Hypericum triquetrifolium (50% ethanol : 50% water) extract (HTE) on apoptosis, cell cycle modulation, and cell cycle arrest in human colon cancer cell line (HCT-116). They postulated that HTE extract possesses a potent therapeutic activity for colon cancer via cell cycle arrest and apoptosis induction. My comments: 1. ATCC recommends using McCoy's 5A Medium for HTC-116 cells. Why did the authors use RMPI-1640? 2. Authors should compare efficacy with a chemotherapeutic agent commonly used in colorectal cancer, such as 5-fluorouracil. 3. In Figure 4 there is no significant difference between Staurosporine and extract concentration at concentration 0 (control)? 4. The authors should present the peaks of gas chromatography-mass spectrometry (GC-MS) in the manuscript.

Author Response

Comments and Suggestions for Authors

In the manuscript, the authors investigated the role of Hypericum triquetrifolium (50% ethanol : 50% water) extract (HTE) on apoptosis, cell cycle modulation, and cell cycle arrest in human colon cancer cell line (HCT-116). They postulated that HTE extract possesses a potent therapeutic activity for colon cancer via cell cycle arrest and apoptosis induction.

My comments:

ATCC recommends using McCoy's 5A Medium for HTC-116 cells. Why did the authors use RMPI-1640?

Both types of mediums are used. This is a link for one paper example used RPMI for HCT-116 (https://www.ncbi.nlm.nih.gov/pmc/articles/PMC4859796/)

Authors should compare efficacy with a chemotherapeutic agent commonly used in colorectal cancer, such as 5-fluorouracil.

We did not include other chemicals in this study as we focused only on chemicals detected in HTE extract. Since HTE was effective in apoptosis induction, cell cycle and caspase cleavage at nontoxic concentrations, we though to proceed in detecting the chemical composition of the extract and focus in their potential mechanism of action. Our aim was to point out potent active compounds in Hypericum triquetrifolium extract and the potential targets in HCT-116 cells.

In Figure 4 there is no significant difference between Staurosporine and extract concentration at concentration 0 (control)?

Staurosporine led to about 100% caspase cleavage. At the control only 15% of the caspase was cleaved.

The authors should present the peaks of gas chromatography-mass spectrometry (GC-MS) in the manuscript.

The GC/MS chromatogram is presented now in a new figure, number 6.
